# Association of Hypernatremia with Immune Profiles and Clinical Outcomes in Adult Intensive Care Unit Patients with Sepsis

**DOI:** 10.3390/biomedicines10092285

**Published:** 2022-09-14

**Authors:** Chiung-Yu Lin, Yu-Mu Chen, Yi-Hsuan Tsai, Kai-Yin Hung, Ying-Tang Fang, Yu-Ping Chang, Meng-Yun Tsai, Hsuan-Feng Wu, Meng-Chih Lin, Wen-Feng Fang

**Affiliations:** 1Division of Pulmonary and Critical Care Medicine, Department of Internal Medicine, Kaohsiung Chang Gung Memorial Hospital, Chang Gung University College of Medicine, Kaohsiung 833, Taiwan; 2Graduate Institute of Clinical Medical Sciences, Chang Gung University, Taoyuan 333, Taiwan; 3Department of Nutritional Therapy, Kaohsiung Chang Gung Memorial Hospital, Kaohsiung 833, Taiwan; 4Department of Respiratory Therapy, Kaohsiung Chang Gung Memorial Hospital, Chang Gung University College of Medicine, Kaohsiung 833, Taiwan; 5Department of Respiratory Care, Chang Gung University of Science and Technology, Chiayi 613, Taiwan

**Keywords:** hypernatremia, granulocyte-colony stimulating factor, tumor necrosis factor α, inflammatory cytokine release downregulation, mortality outcome

## Abstract

Both hypernatremia and an abnormal immune response may increase hospital mortality in patients with sepsis. This study examined the association of hypernatremia with abnormal immune response and mortality in 520 adult patients with sepsis in an intensive care unit (ICU). We compared the mortality and ex vivo lipopolysaccharide (LPS)-induced inflammatory response differences among patients with hyponatremia, eunatremia, and hypernatremia, as well as between patients with acquired hypernatremia on ICU day 3 and those with sustained eunatremia over first three ICU days. Compared with eunatremia or hyponatremia, hypernatremia led to higher 7 day, 14 day, 28 day, and hospital mortality rates (*p* = 0.030, 0.009, 0.010, and 0.033, respectively). Compared with sustained eunatremia, acquired hypernatremia led to higher 7, 14, and 28 day mortality rates (*p* = 0.019, 0.042, and 0.028, respectively). The acquired hypernatremia group nonsignificantly trended toward increased hospital mortality (*p* = 0.056). Day 1 granulocyte colony-stimulating factor (G-CSF) and tumor necrosis factor (TNF) α levels were relatively low in patients with hypernatremia (*p* = 0.020 and 0.010, respectively) but relatively high in patients with acquired hypernatremia (*p* = 0.049 and 0.009, respectively). Thus, in ICU-admitted septic patients, hypernatremia on admission and in ICU-acquired hypernatremia were both associated with higher mortality. The higher mortality in patients with hypernatremia on admission was possibly related to the downregulation of G-CSF and TNF-α secretion after endotoxin stimulation. Compared to sustained eunatremia, acquired hypernatremia showed immunoparalysis at first and then hyperinflammation on day 3.

## 1. Introduction

Hypernatremia is frequently observed in hospitalized patients [1]; the proportion of patients with hypernatremia is generally higher in the intensive care unit (ICU) than in other hospital departments [2,3]. According to a Dutch study, hypernatremia incidence has increased from 13% to 24% over the past two decades [2]. Hypernatremia can cause a series of neurological manifestations, including delirium, stupor, and even coma [4]. It has been reported to be associated with a longer length of mechanical ventilation [5]. More seriously, mortality outcome in septic patients with hypernatremia tends to be higher than in those with eunatremia [6,7].

Many recent studies have focused on abnormal immune responses and the related increase in mortality among septic patients [8,9,10]. We speculate that abnormal immune responses may be the key to increased mortality in hypernatremic sepsis patients. In fact, some studies in the past revealed that hypernatremia may cause inflammation or be associated with inflammation in sepsis [11,12,13]. However, the detailed mechanism remains unclear, such as whether hypernatremia causes immune paralysis or which cytokines participate.

In this combined prospective and retrospective pilot study, we aimed to analyze the hypernatremia–mortality association and determine immune-related differences in ICU patients with hypernatremia. Hence, we compared the mortality outcomes between ICU patients with and without dysnatremia. The effect of ICU-acquired hypernatremia on mortality outcomes was also surveyed. Furthermore, we analyzed the percentage of monocytes expressing HLA-DR to identify whether immune paralysis developed. In addition, we measured cytokine responses in peripheral blood mononuclear cells (PBMCs) following lipopolysaccharide (LPS) stimulation, in order to simulate the cytokine response after exposure to LPS on the outer membrane of Gram-negative bacteria in sepsis.

## 2. Materials and Methods

### 2.1. Patient Enrollment

This study was part of an integrated research program and included a prospective observation study and retrospective medical chart review [14,15,16,17,18,19,20,21,22,23,24]. It was conducted at Kaohsiung Chang Gung Memorial Hospital, a 2700-bed tertiary teaching hospital in southern Taiwan, and included patients with severe sepsis or septic shock who had been admitted to the medical ICU of the hospital between August 2013 and June 2016. The patients were enrolled only if they consented to undergo blood sampling during ICU hospitalization. However, they were excluded if they met any of the following criteria: (1) age <18 years, (2) sepsis developing >24 h before ICU admission, and (3) granulocyte colony-stimulating factor administration ≤1 week before ICU admission.

### 2.2. Study Design

We determined baseline cytokine levels and analyzed cytokine level trends in our patients with hyponatremia, eunatremia, and hypernatremia. Moreover, the 7 day, 14 day 28 day, and hospital mortality rates were compared among patients with hyponatremia, eunatremia, and hypernatremia and then specifically between patients with sustained eunatremia on ICU day 3 and those with hypernatremia acquired on ICU day 3.

Some of the enrolled patients were also participants of another prospective immune-related study. We extracted and compared their cytokine data, including their G-CSF, interleukin (IL)-10, IL-6, and tumor necrosis factor (TNF)-α levels.

The study design was approved by the Institutional Review Board of Chang Gung Memorial Hospital without any conflict of interest.

### 2.3. Definitions

The enrolled patients fulfilled the definition for “severe sepsis” established (for the first time) by the 2001 International Sepsis Definitions Conference and the Surviving Sepsis Campaign [25], that for “sepsis” provided by the Third International Consensus Definitions for Sepsis and Septic Shock (Sepsis-3) [26], or both. For hypernatremia and hyponatremia, we adopted the following definitions: serum sodium concentration of >145 and <135 mEq/L, respectively.

### 2.4. PBMC Preparation

Whole-blood samples (20 mL) were obtained from each patient and stored in a heparin tube (BD, Franklin Lakes, NJ, USA). The first day of blood sampling was identified as day 1. In a Ficoll-Paque tube (Amersham Biosciences, Uppsala, Sweden), whole blood was centrifuged at 400× *g* for 30 min to separate plasma from PBMCs. These PBMCs were collected and analyzed immediately, and the plasma samples were stored at −80 °C until use. The fresh PBMCs were aliquoted into two parts; one was used for HLA-DR-expressing monocyte percentage measurement, and the other was used for cell culture.

### 2.5. HLA-DR-Expressing Monocyte Percentage Measurement through Flow Cytometry

We measured the percentage of HLA-DR-expressing monocytes through flow cytometry on Cytomics FC500 (Beckman Coulter, Fullerton, CA, USA). Staining and cell acquisition were performed within 1 h of blood sample collection. Monoclonal antibodies were used as follows: CD14-PerCP/Cy5.5 (clone: HCD14; Biolegend, San Diego, CA, USA) and HLA-DR FITC (clone: L243; Biolegend) per 100 μL of PBMC blood. Negative controls were mouse monoclonal antibodies IgG1 per CP/Cy5.5 (MOPC-21), IgG2a FITC (MOPC-173), and IgG2a PE (MOPC-173), all of which were isotype-matched according to the manufacturer’s recommendations. The monocytes were identified on the basis of their CD14 expression. A minimum of 30,000 PBMCs were analyzed from each sample; the results are expressed as percentages of HLA-DR-positive monocytes in the total monocyte population.

### 2.6. Cell Culture and LPS-Stimulated Cytokine Response

PBMCs were added at a density of 1 × 10^6^ to a 5 mL round-bottom polystyrene test tube (BD Falcon, Bedford, MA USA) in which 2 mL of sterile Dulbecco’s modified Eagle’s medium (Gibco, Grand Island, NY, USA) containing 1% heat-inactivated fetal bovine serum (Gibco), 1 mM L-glutamine, and 1 mM sodium pyruvate was present. Inflammation was induced using 100 ng/mL LPS (Sigma, St. Louis, MO, USA). All tubes were incubated at 37 °C in 5% CO_2_ for 4 h. Samples of the conditioned media were analyzed for cytokine expression levels.

The LPS-stimulated cytokine response was considered to be the fold elevation in the level of cytokines released by stimulated PBMCs in the conditioned media. It was, thus, calculated by dividing the cytokine level after LPS stimulation with that before stimulation.

### 2.7. Milliplex Assay

We next quantified levels of G-CSF, IL-10, IL-6, and TNF-α in the conditioned media using a Human Cytokine/Chemokine Magnetic Bead Panel-equipped Milliplex MAP kit (#HCYTOMAG-60K, EMD Millipore, Darmstadt, Germany) according to the manufacturer’s instructions. The standards and samples were analyzed on a MAGPIX System device (Millipore) using the software program MILLIPLEX Analyst 5.1 with a five-parameter logistic curve-fitting model.

### 2.8. Statistical Analyses

Statistical analysis was performed using SPSS (version 21.0; IBM Corp., Armonk, NY, USA). Categorical variables were compared using the chi-square test or Fisher’s exact test as appropriate, whereas continuous variables were analyzed using Student’s *t*-test or Mann–Whitney U test as appropriate. Multivariate analysis for independent prognostic factor selection was performed using logistic regression. The Kaplan–Meier estimator and the log-rank test were used to determine the effects of the different immune dysfunction scores on patient survival. The Kruskal–Wallis test was used to determine the correlation of LPS-stimulated cytokine release and HLA-DR-expressing monocyte percentage with dysnatremia. A *p*-value < 0.05 was considered to indicate statistical significance.

## 3. Results

### 3.1. Patient Characteristics

Of the 2744 patients admitted to the ICU of Kaohsiung Chang Gung Memorial Hospital between August 2013 and June 2016, 520 sepsis patients with available baseline serum sodium data were included in our final analysis (Figure 1). Of these patients, 164 (31.5%) had hyponatremia, 51 (9.8%) had hypernatremia, and 305 (58.7%) had eunatremia.

### 3.2. Baseline Clinical Parameters and Immune Profiles of Patients with Sepsis

Patients with hypernatremia were considerably older than patients with eunatremia or hyponatremia (*p* = 0.036; Table 1). However, the differences in body mass index, sex, history of diabetes mellitus, hypertension, coronary artery disease, chronic obstructive airways disease, cirrhosis, stroke, chronic renal disease, and baseline Acute Physiology and Chronic Health Evaluation (APACHE II) scores between patients with and without dysnatremia were nonsignificant.

Baseline plasma cytokine levels of only 160 of the 520 included patients were available. However, the differences in baseline IL-6, IL-10, G-CSF, and TNF-α levels between patients with and without dysnatremia were nonsignificant (Appendix A)

### 3.3. Effects of Dysnatremia on Clinical Outcomes

All 520 patients were divided into three groups on the basis of their sodium levels on ICU day 1: hyponatremia (*n* = 164), eunatremia (*n* = 305), and hypernatremia (*n* = 51). All three groups demonstrated significant differences in 7 day, 14 day, 28 day, and hospital mortality rates (*p* = 0.030, 0.009, 0.010, and 0.033, respectively; Table 2). Specifically, compared with the eunatremia group, the hypernatremia group demonstrated higher 7 day, 14 day, 28 day, and hospital mortality rates (*p* = 0.014, 0.004, 0.003, and 0.014, respectively). Although the Kaplan–Meier hospital survival curve revealed no significant differences (overall *p* = 0.251), the hypernatremia group demonstrated a significant difference in survival compared with the hyponatremia (*p* = 0.003) and eunatremia (*p* = 0.029) groups (Figure 2a).

Of all the patients with eunatremia on ICU day 1, we divided those with sustained eunatremia on ICU day 3 (*n* = 197) and those with acquired hypernatremia on ICU day 3 (*n* = 52) into two groups. The acquired hypernatremia group had higher 7, 14, and 28 day mortality rates (*p* = 0.019, 0.042, and 0.028, respectively; Table 2). Although nonsignificant, patients with acquired hypernatremia exhibited a trend toward a high mortality rate (*p* = 0.056). However, the Kaplan–Meier curve revealed nonsignificant differences in hospital survival (*p* = 0.009; Figure 2b) (Appendix A).

The multivariate analysis using a binary logistic regression model confirmed that hypernatremia is an independent risk factor for hospital mortality (odds ratio = 2.300; 95% confidence interval = 1.179–4.487; *p* = 0.015; Table 3).

### 3.4. LPS-Stimulated Cytokine Release and HLA-DR-Expressing Monocyte Percentage

The differences in day 1 G-CSF and TNF-α response after LPS stimulation among patients with hyponatremia, eunatremia, and hypernatremia were significant (*p* = 0.020 and 0.010, respectively). The hypernatremia group demonstrated higher day 1 G-CSF and TNF-α responses than the eunatremia group (*p* = 0.019 and 0.046, respectively; Figure 3a). Moreover, the differences in LPS-stimulated cytokine release among patients with hyponatremia, eunatremia, and hypernatremia in day 3 were nonsignificant (Figure 3b).

Patients with sustained eunatremia and those with acquired hypernatremia demonstrated nonsignificant differences in day 1 LPS-stimulated cytokine release (Figure 3c). However, compared with patients with sustained eunatremia, those with acquired hypernatremia had significantly higher day 3 G-CSF and TNF-α responses (*p* = 0.049 and 0.009, respectively; Figure 3d).

Day 1 HLA-DR-expressing monocyte percentages demonstrated significant differences among the hyponatremia, eunatremia, and hypernatremia groups (*p* = 0.025); moreover, the hypernatremia group demonstrated the lowest mean HLA-DR-expressing monocyte percentage (*p* (hypernatremia vs. hyponatremia) = 0.050; Figure 3e). However, day 1 or day 3 HLA-DR-expressing monocyte percentages revealed nonsignificant differences between patients with sustained eunatremia and those with acquired hypernatremia (Figure 3f). Medians and interquartile ranges of each cytokine response and HLA-DR-expressing monocyte percentage are listed in Table 4.

### 3.5. Validation Using a Validation Cohort

Next, we validated our prediction model. The validation cohort database contained 515 adult patients with sepsis in ICUs recruited prospectively between December 2019 and March 2021 in Kaohsiung Chang Gung Memorial Hospital. In this cohort, sepsis was defined using the definition of Sepsis-3. Immune cytokines data were unavailable for this validation cohort database.

The validation cohort included 187, 280, and 48 patients with hyponatremia, eunatremia, and hypernatremia, respectively. The differences in 7, 14, and 28 day mortality rates among these three groups were nonsignificant (*p* = 0.870, 0.725, and 0.786, respectively) but that of hospital mortality rate was significant (*p* = 0.019). Specifically, patients with hypernatremia demonstrated the highest mortality rate (54.2%), followed by those with hyponatremia (38.0%) and then those with eunatremia (33.2%).

The validation cohort included 187 patients with sustained eunatremia and 44 patients with acquired hypernatremia. Compared with patients with sustained eunatremia, those with acquired hypernatremia had nonsignificantly higher 7 day (2.7% vs. 9.1%), 14 day (6.4% vs. 25.0%), 28 day (15.5% vs. 31.8%), and hospital (28.3% vs. 50.0%) mortality rates (*p* = 0.070, 0.001, 0.018, and 0.007, respectively).

## 4. Discussion

Hypernatremia affects many aspects of clinical outcome. Hypernatremia can cause neurological manifestations [4] and is associated with multiple systemic adverse effects, including increased insulin resistance [27], decreased lactate clearance [28], impaired left-ventricular contractility [29], and prolonged mechanical ventilation days [30]. Moreover, hypernatremia at ICU admission has been reported to be an independent risk factor for poor sepsis prognosis [31]. Acquired hypernatremia after ICU admission is also associated with a higher mortality rate [32]. Our study revealed that hypernatremia, whether developed upon ICU admission or acquired after ICU admission, can worsen sepsis prognosis, possibly due to immune paralysis.

In patients with sepsis, hypernatremia etiology is complex and remains unclear; it cannot be completely explained by inappropriate fluid balance or sodium intake alone [33,34]. We analyzed the fluid input or input–output profiles of different groups in the first 3 days in our study. Although there was no statistically significant difference (Appendix A), a trend of higher fluid input presented on the first ICU day in hypernatremia group (hyponatremia versus eunatremia versus hypernatremia: 1703.7 mL versus 1735.8 mL versus 2034.2 mL; *p* = 0.066). Here, we do not deny the influence of excessive infusion on serum sodium; we believe that the causes of hypernatremia in sepsis are complex but go beyond excessive infusion only. Clinicians need to consider different factors to explore the causes driving hypernatremia, e.g., drugs or insensible water loss.

Recent evidence has linked hypernatremia to the persistent inflammation and immune suppression in septic patients [13]. Understanding the relationship between hypernatremia and inflammation in sepsis can provide a clearer picture of the high mortality causes. According to the literature, sodium stored in local tissue, such as skin, plays an important role in inflammation and defense against infection [35]. Skin infection drives local salt accumulation, and the skin tissue sodium levels reduce upon antibiotic treatment [36]. Infection protection mechanisms may include both osmotic protection and inflammatory responses. Experts studied this phenomenon and simulated the inflammatory response under high salt conditions with LPS stimulation in vitro. The result showed that high salt condition, with a 40 mM increase in culture medium NaCl concentration, augments LPS-mediated and IL-1α or IL-1β + TNF-induced macrophage activation [36]. Therefore, acquired hypernatremia may cause hyperinflammation.

In addition, there is currently other evidence that high salt can affect the adaptive immunity, such as activating T-cell proliferation, which forms an essential part of the antigen-specific adaptive immune system [37,38]. Available studies suggest that high-salt conditions favor T-cell proliferation and drive these cells toward a proinflammatory phenotype, while impairing the tolerogenic function of these cells [35]. T cells differentiate into cytotoxic, helper, and regulatory T cells. High salt conditions boost the development of IL-17-producing CD4^+^ T helper cells (Th17 cells) specifically via NFAT5- and serum/glucocorticoid-regulated kinase 1 (SGK1)-dependent signaling. [39,40] The studies addressing the effects of high salt on B cells are currently limited. One reported that increased osmolality boosts B-cell activation and differentiation, but increased cell death and impaired plasma blast differentiation were also observed after 72 h exposure to high-salt conditions [13,41].

In the current study, day 1 G-CSF and TNF-α responses after LPS stimulation were significantly lower in the hypernatremia group than in the eunatremia group. Therefore, G-CSF and TNF-α release might be downregulated in patients with hypernatremia. G-CSF, a proinflammatory cytokine, stimulates the bone marrow to produce neutrophils and aids in mobilizing hematopoietic stem cells from the bone marrow into the blood [34]. Moreover, TNF-α, another proinflammatory cytokine produced by activated macrophages and monocytes, can enhance macrophage production from progenitor cells and promote macrophage activation and differentiation [42,43]. Decreased TNF-α secretion after LPS stimulation has also been noted in sepsis non-survivors [8]. Because the hypernatremia group demonstrated G-CSF and TNF-α downregulation, as well as a lower HLA-DR-expressing monocyte percentage, the potential for immune paralysis should be considered. With regard to the ICU-acquired hypernatremia, an upregulation of day 3 G-CSF and TNF-α release was observed, suggesting that patients with acquired hypernatremia demonstrate a hyperinflammatory status. 

The current study revealed no correlation between dysnatremia and LPS-stimulated IL-6 or IL-10 response. Thus far, only a few studies have discussed the relationship of hypernatremia with IL-6 and IL-10. In one study, enhanced IL-6 release was observed after LPS stimulation after preincubation of PBMCs with glucose (1000 mg/dL) or mannitol (1000 mg/dL) [44]. A mouse study revealed significant increases in IL-6 in intestinal epithelial cells exposed to 627 mOsm dextran sulfate sodium solution [45]. However, the hyperosmotic stress in both the aforementioned studies was not induced by hypernatremia. By contrast, an ex vivo human whole blood study reported that increased IL-6, TNF-α, and IL-10 were observed after LPS-induced inflammation in a high-salt medium [46]. In that study, the authors concluded that salt could increase monocyte CCR2 expression and inflammatory responses in humans.

Our subsequent validation cohort study revealed a significant association of hypernatremia or acquired hypernatremia with hospital mortality. Both hypernatremia and acquired hypernatremia were not associated with the 7 day mortality, while hypernatremia and 14 or 28 day mortality rate were nonsignificantly associated. These validation results indicated that hypernatremia is associated with hospital mortality rate, but the association of hypernatremia with short-term mortality remains under debate. In practice, clinicians should consider multiple aspects to predict mortality rather than relying solely on a single risk factor.

Our current results reconfirmed the relationship between hypernatremia and mortality. We also made some findings regarding cytokine responses in hypernatremia. Significant differences in TNF-α and G-CSF between hypernatremia patients and other groups are evidence that hypernatremia is associated with inflammation in sepsis. However, we cannot tell from this article whether inflammation causes hypernatremia, whether hypernatremia worsens inflammation, or both. We also cannot know whether prolonged hypernatremia increases mortality from sepsis. Regardless, the role of hypernatremia in sepsis has recently become increasingly accepted; the use of balanced crystalloids or albumin instead of 0.9% saline during fluid resuscitation in sepsis is also more appreciated [47]. 

The results of this article prompt a follow-up question: Should we actively correct the hypernatremia in septic patients after volume-resuscitated shock? Although one article reported that not resuscitating severe hypernatremia (serum sodium ≥155 mEq/L) in the first 24 h could lead to higher 30 day mortality [48], it was retrospectively designed and did not address septic patients in the ICU. To our knowledge, there are no large and well-designed studies discussing active hypernatremia correction in shock or septic patients [49]. There are also no globally recognized guidelines for the hypernatremia correction strategies in sepsis. How severe should the hypernatremia be to initiate correction, and what range should the correction rate be controlled within? These are important things that clinicians need to think about since an inappropriate serum sodium correction might disrupt body fluid balance and worsen the burden on the renal and cardiopulmonary system in septic patients.

In this article, we looked at the immune status of different groups in two ways: by measuring HLA-DR to see if immunosuppression occurred, and by stimulating PBMCs with lipopolysaccharide to simulate the cytokine response in sepsis; we analyzed both day 1 and day 3 data in the ICU. This is one of the strengths of our article. We also validated the relationship between mortality and serum sodium in the article, which is another strength of the article. To our knowledge, only a few studies have explored these inflammatory responses in septic patients with hypernatremia and acquired hypernatremia. Here, we discussed both hypernatremia on admission and ICU-acquired hypernatremia.

Compared with two other studies with larger case numbers presented by Cecilia Chi et al. (>155 mEq/L: 35.3%) [50] and Amit Akirov et al. (>150 mEq/L: 52.2%) [51], our study revealed a higher hospital mortality rate in patients with hypernatremia on admission (>145 mEq/L: 58.8%). The differences in the ICU patient severity, medical equipment, or treatment modalities may have led to the differences in mortality rates. Another important reason is that our study focused on “septic patients” in the ICU, whereas the other two articles considered patients in ICUs and medical wards, respectively. Another recent prospective observational study in a single center, similarly focused on ICU sepsis patients [6], had a similar 7 day mortality rate (>145 mEq/L: 29.3%) to ours (>145 mEq/L: 27.5%). That article analyzed the 7 day mortality and length of ICU stay, rather than hospital mortality. As far we know, mortality rates in our article were not found to be significantly different from other studies under comparative conditions.

This pilot study had several limitations owing to its design. Our data were extracted from “Internal Medicine” ICU patients over the age of 18 with severe sepsis (or sepsis defined by Sepsis-3 criteria) or septic shock; therefore, the current results may not be generalizable to all types of ICUs or patients. Furthermore, the cytokine data were available only for some of the included patients, and no cytokine data could be extracted in the validation cohort. Validation would be even more important in studies with small case numbers; moreover, the case number also limited the further subsequent group analysis (e.g., the cytokine response between mild hypernatremia and severe hypernatremia). A larger study with a high case number and well-documented experimental design to confirm the association of hypernatremia with inflammatory cytokines is warranted. Lastly, similar to problems faced by other ex vivo trials, the ex vivo LPS-stimulated response might not be totally representative of the cytokine presentation in real septic human beings. 

## 5. Conclusions

Hypernatremia on admission is associated with increased hospital mortality and abnormal immune response in sepsis, as demonstrated by the downregulation of G-CSF and TNF-α release after LPS stimulation of cells ex vivo. Moreover, hypernatremia acquired in the ICU can increase hospital mortality, probably linked to immunoparalysis, and then hyperinflammation on day 3 compared to the sustained eunatremia.

## Figures and Tables

**Figure 1 biomedicines-10-02285-f001:**
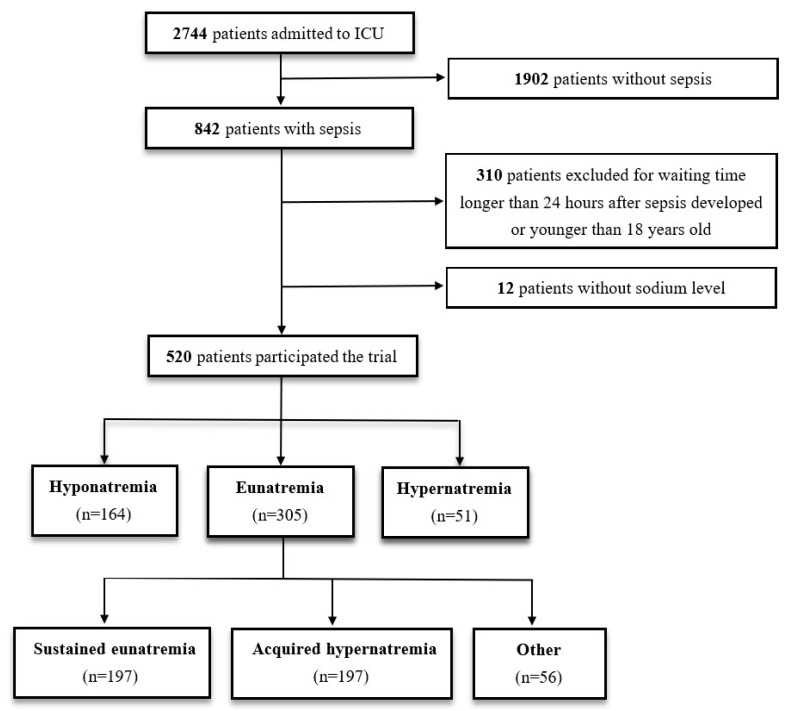
Patient inclusion and assignment. ICU, intensive care unit.

**Figure 2 biomedicines-10-02285-f002:**
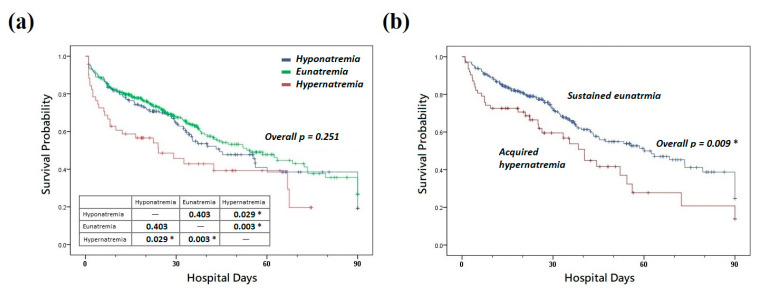
Kaplan–Meier survival curves for (**a**) patients with hyponatremia, eunatremia, and hypernatremia and (**b**) patients with sustained eunatremia and acquired hypernatremia. * *p* < 0.05.

**Figure 3 biomedicines-10-02285-f003:**
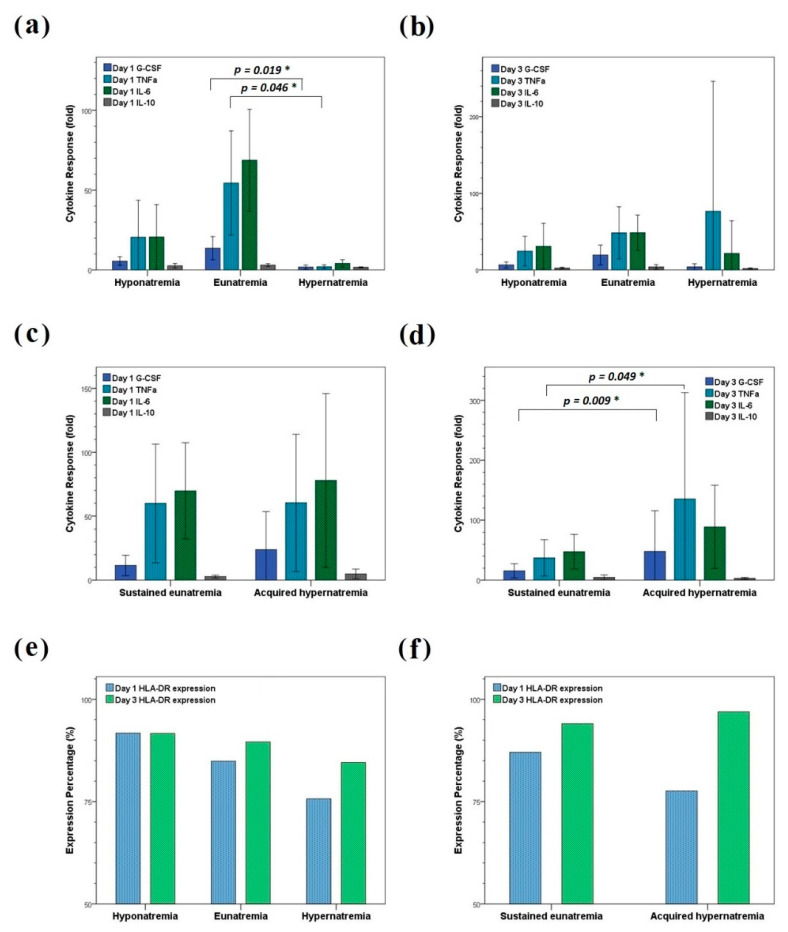
LPS-stimulated cytokine response and HLA-DR-expressing monocyte percentage. (**a**–**d**) Bar charts for LPS-stimulated cytokine response (unit: fold) in patients with hyponatremia, eunatremia, and hypernatremia on ICU days 1 (**a**) and 3 (**b**) and in patients with sustained eunatremia and acquired hypernatremia on days 1 (**c**) and 3 (**d**). Error bars represent 95% confidence intervals. The LPS-stimulated cytokine response was calculated by dividing the cytokine level after LPS stimulation with that before stimulation. (**e**,**f**) Bar charts of day 1 and 3 HLA-DR-expressing monocyte percentage in patients with hyponatremia, eunatremia, and hypernatremia (**e**) and in patients with sustained eunatremia and acquired hypernatremia (**f**). Abbreviations: IL, interleukin; LPS: lipopolysaccharide; TNF-α: tumor necrosis factor-α; G-CSF, granulocyte colony-stimulating factor. * *p* < 0.05.

**Table 1 biomedicines-10-02285-t001:** Baseline clinical parameters and plasma cytokine levels in patients with and without dysnatremia on admission.

Clinical Parameters	All (*n* = 520)	Hyponatremia (*n* = 164)	Eunatremia (*n* = 305)	Hypernatremia (*n* = 51)	*p*-Value
Age, mean (SD)	66.6 (15.3)	66.2 (14.9)	65.9 (15.7)	71.8 (12.9)	0.036 *
BMI, mean (SD)	22.8 (5.0)	22.8 (4.8)	22.8 (5.1)	22.4 (4.5)	0.912
Gender (male), *n* (%)	212 (40.8)	77 (47.0)	112 (36.7)	23 (45.1)	0.080
Diabetes mellitus, *n* (%)	232 (44.6)	81 (49.4)	128 (42.0)	23 (45.1)	0.304
Hypertension, *n* (%)	276 (53.1)	87 (53.0)	160 (52.5)	29 (56.9)	0.844
CAD, *n* (%)	130 (25.0)	38 (23.2)	78 (25.6)	14 (27.5)	0.775
COPD, *n* (%)	62 (11.9)	14 (8.5)	39 (12.8)	9 (17.6)	0.165
Cirrhosis, *n* (%)	40 (7.7)	8 (4.9)	28 (9.2)	4 (7.8)	0.249
Stroke, *n* (%)	109 (21.0)	31 (18.9)	64 (21.0)	14 (27.5)	0.424
CKD, *n* (%)	138 (26.5)	41 (25.0)	85 (27.9)	12 (23.5)	0.700
APCHE II score, mean (SD)	25.0 (8.8)	24.3 (8.6)	24.8 (8.8)	27.6 (9.1)	0.088

Abbreviations: BMI, body mass index; CAD, coronary artery disease; CKD, chronic kidney disease; COPD, chronic obstructive pulmonary disease; OI, oxygenation index; APACHE, Acute Physiology and Chronic Health Evaluation; SD, standard deviation. * *p* < 0.05.

**Table 2 biomedicines-10-02285-t002:** Mortality outcomes of patients with sepsis.

	All (*n* = 520)	Hyponatremia (*n* = 164)	Eunatremia (*n* = 305)		Hypernatremia (*n* = 51)	*p*-Value
7 day mortality, *n* (%)	77 (14.8)	22 (13.4)	41 (13.4)		14 (27.5)	0.030 *
14 day mortality, *n* (%)	115 (22.1)	34 (20.7)	61 (20.0)		20 (39.2)	0.009 *
28 day mortality, *n* (%)	153 (29.4)	49 (29.9)	80 (26.2)		24 (47.1)	0.010 *
Hospital mortality, *n* (%)	225 (43.3)	74 (45.1)	121 (39.7)		30 (58.8)	0.033 *
	**All** **(*n* = 249)**	**Sustained Eunatremia** **(*n* = 197)**		**Acquired Hypernatremia** **(*n* = 52)**		** *p* ** **-Value**
7 day mortality, *n* (%)	25 (10.0)	15 (7.6)		10 (19.2)		0.019 *
14 day mortality, *n* (%)	43 (17.3)	29 (14.7)		14 (26.9)		0.042 *
28 day mortality, *n* (%)	60 (24.1)	41 (20.8)		19 (36.5)		0.028 *
Hospital mortality, *n* (%)	99 (39.8)	72 (36.5)		27 (51.9)		0.056

The 7 day (*p* = 0.014), 14 day (*p* = 0.004), 28 day (*p* = 0.003), and hospital (*p* = 0.014) mortality rates were significantly higher in the hypernatremia group than in the eunatremia group. * *p* < 0.05.

**Table 3 biomedicines-10-02285-t003:** Multivariate binary regression analysis for hospital mortality risk factors.

Parameter	Odds Ratio	95% Confidence Interval	*p*-Value
Age	1.001	0.986–1.016	0.882
BMI	0.957	0.916–1.000	0.048 *
Gender (male)	0.956	0.634–1.441	0.830
Diabetes mellitus	0.814	0.533–1.242	0.340
Hypertension	0.823	0.524–1.293	0.399
CAD	1.186	0.725–1.941	0.496
COPD	0.996	0.543–1.828	0.991
Cirrhosis	1.441	0.680–3.053	0.340
Stroke	0.686	0.405–1.162	0.161
CKD	1.227	0.771–1.954	0.388
APCHE II score	1.037	1.014–1.060	0.002 *
Hypernatremia	2.300	1.179–4.487	0.015 *

Abbreviations: BMI, body mass index; CAD, coronary artery disease; CKD, chronic kidney disease; COPD, chronic obstructive pulmonary disease; OI, oxygenation index; APACHE, Acute Physiology and Chronic Health Evaluation. * The parameter “hypernatremia” represents patients with hypernatremia on admission versus all patients without hypernatremia but with eunatremia or hyponatremia on admission. * *p* < 0.05.

**Table 4 biomedicines-10-02285-t004:** Immune stimulation response and HLA-DR-expressing monocyte percentage in patients with sepsis.

	All	Hyponatremia	Eunatremia		Hypernatremia	*p*-Value
**Day 1**	(*n* = 128)	(*n* = 42)	(*n* = 75)		(*n* = 11)	
G-CSF	2.4 (4.8)	2.4 (5.6)	2.6 (4.7)		1.0 (2.6)	0.020 *
TNF-α	2.6 (6.0)	2.1 (4.8)	2.8 (28.2)		1.9 (3.1)	0.010 *
IL-6	3.4 (18.8)	2.8 (6.6)	3.9 (62.4)		2.8 (2.6)	0.071
IL-10	1.7 (1.0)	1.7 (1.1)	1.6 (1.2)		1.4 (0.9)	0.758
HLA-DR expression, %	92.2 (15.0)	95.5 (8.2)	90.0 (18.9)		87.3 (35.4)	0.025 *
**Day 3**	(*n* = 115)	(*n* = 39)	(*n* = 67)		(*n* = 9)	
G-CSF	2.4 (4.4)	2.3 (4.9)	2.6 (3.9)		1.0 (3.4)	0.899
TNF-α	2.3 (4.5)	2.0 (4.5)	2.7 (28.2)		1.9 (2.4)	0.724
IL-6	3.2 (12.5)	2.8 (5.8)	3.6 (62.4)		2.8 (2.4)	0.729
IL-10	1.7 (0.9)	1.7 (1.0)	1.6 (1.0)		1.4 (0.8)	0.910
HLA-DR expression, %	92.4 (16.0)	95.6 (7.9)	90.0 (19.4)		87.3 (35.8)	0.336
	**All**	**Sustained Eunatremia**		**Acquired Hypernatremia**		** *p* ** **-Value**
**Day 1**	(*n* = 60)	(*n* = 49)		(*n* = 11)		
G-CSF	2.6 (6.0)	2.4 (3.8)		4.7 (21.6)		0.184
TNF-α	2.8 (51.5)	2.6 (28.4)		16.8 (95.4)		0.248
IL-6	4.3 (83.8)	3.7 (81.3)		59.2 (103.4)		0.344
IL-10	1.6 (1.2)	1.6 (0.9)		2.2 (5.9)		0.240
HLA-DR expression, %	89.9 (20.9)	90.5 (20.2)		81.5 (20.3)		0.136
**Day 3**	(*n* = 58)	(*n* = 47)		(*n* = 11)		
G-CSF	2.6 (4.8)	2.3 (3.8)		4.7 (21.6)		0.049 *
TNF-α	2.7 (4.4)	2.6 (28.4)		16.8 (95.4)		0.009 *
IL-6	3.8 (78.7)	3.6 (76.3)		59.2 (103.4)		0.077
IL-10	1.6 (1.1)	1.5 (0.8)		2.2 (5.9)		0.293
HLA-DR expression, %	90.0 (21.3)	90.8 (20.8)		81.5 (20.3)		0.413

Abbreviations: G-CSF, granulocyte colony-stimulating factor; IL, interleukin; LPS, lipopolysaccharide; TNF, tumor necrosis factor; HLA, human leukocyte antigen; IQR, interquartile range. All data are presented as medians (IQRs). Day 1 G-CSF and TNF-α levels were higher in the eunatremia group than in the hypernatremia group (*p* = 0.019 and 0.046, respectively); HLA-DR-expressing monocyte percentage was higher in the hypernatremia group than in the hyponatremia group (*p* = 0.050). * *p* < 0.05

## Data Availability

Information related to this study can be obtained by emailing the authors.

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
