# Peer review of "Association of Hypernatremia with Immune Profiles and Clinical Outcomes in Adult Intensive Care Unit Patients with Sepsis"

_biomedicines, 2022, doi:10.3390/biomedicines10092285_

Round 1

Reviewer 1 Report

The manuscript is interesting and, in agreement with the issue. Some changes are required to improve the manuscript.

a) The entire manuscript should be revised to avoid some grammar mistakes. Also, a pair of repited words (together).

b) Please highlight the results with significance (with an asterisk or a symbol to easy identification in tables and figures). 

c) I suggest you to change "lipopolysaccharide (LPS)-induced inflammatory response differences" to "ex vivo lipopolysaccharide (LPS)-induced inflammatory response differences".

d) The discussion section could be enriched by comparisson with additional previous works. In fact, just 7 references are added, and many others used as in introduction. However, multiple recent approaches have been reported in the sepsis+inflammatory diagnosis field.

e) At the end of discussion, limitations could be enriched. Particular attention should be to be clear regarding the data are collected in a hospital with 'x,y and z' characteristics; then, additional comparison with similar studies in other hospitals and conditions around the world could be appreciated. 

f) in conclusions: Is adequate to sentence?: Hypernatremia on admission is associated with increased hospital mortality and abnormal immune response in sepsis, as demonstrated by the downregulation of G-CSF and 349 TNF-α release after LPS stimulation of cells ex vivo. Moreover, hypernatremia acquired in the ICU can increase hospital mortality (probably linked to this abnormal response).

Reviewer 2 Report

The study described in the paper deals with the evaluation of clinical and immunological effects hypernatremia in patients admitted at the ISU with sepsis. Data confirmed the known association of unfavorable outcome in ICU patients with the increased concentration of sodium in plasma. The strength of the paper is demonstrated by a new, clinically validated finding that the increased sodium concentration early on admittance associates with both hospital mortality and suppression of immune responses (HLA-DR decrease on monocytes. Another strength includes the association of proinflammatory reaction (increased TNF release on day 3 after hospitalization) and acquired hypernatremia. Some weakness comes from insufficient explanation of the mechanisms of the phenomenon (presumably, immune paralysis, or exhaustion versus hyperinflammatory condition) that may origin from insufficient proper citations of papers directly related to the study focus. Improving discussion of papers (De Freitas et al, 2019; Akirov et al., 2017Chi et al., 2021) that have demonstrated association of increased sodium in plasma and  mortality in critical illness. Further, the discussion of the most relevant to the subject paper that links sodium increase and regulation of immunity (as an  example, a review by Shatz V.  et al., 2016, not included in a list of references) might definitely  benefit the final version of the paper. Th references should be checked again for proper citation: there is a misspelling of the last name of the same first author in refs 5 and 33 (the real name is van de Louw,  not "A").  

Round 2

Reviewer 1 Report

The authors addressed all comments and suggestions. The manuscript is improved and now it can be suitable to the special issue in biomedicines.